



# The role of air-sea heat fluxes for marine heatwaves in the Mediterranean Sea

Dimitra Denaxa[1,2], Gerasimos Korres[1], Giulia Bonino[3], Simona Masina[3], Maria Hatzaki[2]

[1] Hellenic Centre for Marine Research (HCMR), Greece
[2] National and Kapodistrian University of Athens, Department of Geology and Geoenvironment, Greece
[3] Centro Euro-Mediterraneo sui Cambiamenti Climatici (CMCC), Italy

*Correspondence to*: Dimitra Denaxa (ddenaxa@hcmr.gr)

**Abstract.**

Recent research studies have significantly contributed in understanding physical mechanisms associated with the occurrence of Marine Heatwaves (MHWs). Building upon prior research, this study investigates the relative role of air-sea heat exchange and oceanic processes during the onset and decline phases of surface MHWs in the Mediterranean Sea, based on a joint analysis of remote sensing data and reanalysis outputs over the period 1993-2022. Our findings suggest that oceanic processes play a key role in driving SST anomalies during MHWs, as 44% of the onset and only 17% of the decline phases are primarily driven by surface heat fluxes. The role of surface fluxes becomes more important during warmer months and onset periods. Spatially, their contribution is greater in the Adriatic and Aegean sub-basins, where they become the major driver of most onset phases. Latent heat emerges as the most significant heat flux component in forming the SST evolution, across all seasons. Onset/decline phases lasting less than 5 days experience a weaker contribution of heat fluxes, compared to longer phases (lasting 5-10 or more than 10 days). Moreover, an inverse relationship between MHW severity and the contribution of heat fluxes is observed. At the subsurface, mixed layer shoaling is found over the entire duration of most MHWs, particularly for those of shorter duration. Therefore, the surface cooling right after the peak day is likely not associated with vertical mixing in such cases. These findings suggest that other oceanic processes, potentially horizontal advection, have key role in modulating SST at the beginning of most MHW declines. In turn, further dissipation of heat is commonly driven by vertical mixing, as indicated by a significant mixed layer deepening after the MHW end day in most cases. This study emphasizes the need for considering subsurface information for future studies of MHWs and highlights the importance of accounting for limitations associated with the definitions employed for MHW phases.

## 1. Introduction

Marine heatwaves (MHWs) are extreme events, characterized by prolonged periods of anomalously high water temperature, lasting for at least five consecutive days (Hobday et al., 2016). These events have gathered increased attention due to their detrimental effects on marine life, especially given the increase observed in their frequency, intensity and duration over the recent decades, at global and Mediterranean scale (Oliver et al., 2018, Holbrook et al., 2019; Darmaraki et al., 2019a; Juza et al., 2022; Dayan et al., 2023, Pastor and Khodayar, 2023). Mass mortality events and local extinctions, coral bleaching and massive shifts of marine species




have been extensively reported (Wernberg et al., 2016; Frölicher and Laufkotter, 2018; Smale et al., 2019; Garrabou et al., 2022; Smith et al., 2023), along with socioeconomic impacts on fishery and aquaculture industries (Mills K.E., et al., 2013; Cavole L., 2016). The intensification of MHW conditions has been attributed mostly to ocean warming (Oliver et al., 2018, Ciappa, 2022), while further intensification is expected in the future (Oliver et al., 2019; Darmaraki et al., 2019b; Plecha & Soares, 2019; Hayashida

et al., 2020), driven by anthropogenic forcing and particularly pronounced under high-emission future scenarios (Oliver et al., 2019).

Given these concerns, it is crucial to enhance our understanding of the driving factors behind MHWs at regional scale, particularly within the framework of exploring predictability options and facilitating marine decision-making (Holbrook et al., 2020; Spillman et al., 2021). Recent research has significantly contributed in identifying physical drivers and MHW-favoring conditions (e.g.,

Holbrook et al., 2019; Sen Gupta et al., 2020; Oliver et al., 2021; Vogt et al., 2022; Marin et al., 2022). Individual events in the Mediterranean Sea have also been explored, such as the widely known MHW in 2003 in the western Mediterranean Sea (e.g., Sparnocchia et al., 2006; Olita et al. 2007; Bonino et al., 2023), the short-lasting, record-breaking MHW in May 2020 in the southeastern Mediterranean Sea (Ibrahim et al., 2021; Denaxa et al., 2022) and the most recent long-lasting MHW in summer 2022 (McAdam et al., 2023; Pirro et al., 2023).


However, a limited number of studies have assessed physical drivers separately for the buildup and decay of MHW events, employing different methodologies and datasets. Schlegel et al. (2021) demonstrated that nearly 50% of surface MHWs in the northwest Atlantic are heat flux-driven but less than 20% decay due to heat fluxes, suggesting that oceanic processes are mainly responsible for the MHW decline. Marin et al. (2022) investigated upper-ocean MHWs based on global ocean circulation model

output and found that heat advection, followed by anomalous air-sea heat flux, explain most of the upper ocean temperature anomalies during both MHW onset and decline phases. Moreover, Darmaraki et al. (2023) reports that air-sea fluxes, wind forcing and vertical mixing have a key role during MHWs in the Mediterranean Sea, while horizontal advection appears to dominate at local scales, as suggested by preliminary results from a fully-coupled regional climate system model. Within this context, the present study utilizes high resolution observational Sea Surface Temperature (SST) and modelled heat flux data to assess the driving role

of air-sea heat exchange during onset and decline phases of surface MHWs in the Mediterranean Sea. Furthermore, it provides insights into the concurrent subsurface conditions by examining the mixed layer dynamics during MHWs.

## 2. Data and Methods

MHWs in this study were identified based on high resolution gridded satellite SST data in the Mediterranean Sea. Daily SST values from the Reprocessed and NRT CMEMS datasets (products ref. no. 01 and 02 (Table 1)) were used to cover the period Jan 1993 -

Dec 2022. To study the net surface heat budget ($Q_{net}$), turbulent and radiative surface fluxes were obtained from the ECMWF ERA5 Reanalysis dataset at hourly frequency and 0.25 °x 025 ° horizontal resolution (product ref. no. 05 (Table 1)). Finally, daily values of mixed layer depth (MLD) for the period Jan 1993 - Dec 2022 were obtained from the Mediterranean Sea Physics Reanalysis and



the Mediterranean Sea Physics Analysis and Forecast (products ref. no. 03 and no. 04 (Table 1)). Collocation of SST and MLD with the coarser ERA5 data was performed by using the nearest neighboring value to each ERA5 grid point. The paired values at daily

frequency were used in the heat budget analysis.

MHW detection was performed based on the definition and detection methodology of Hobday et al. (2016), using the matlab toolbox provided by Zhao and Marin (2019). The reference period used in this study to create the daily climatology required for the event detection is the same as the 30-year study period (1993-2022). For the computation of the daily threshold time-series, the 90[th]

percentile was selected, being widely used in MHW studies, thus allowing for a more direct inter-comparison with literature.

Basic properties were computed for each identified event (e.g., start/end day, mean/max intensity, duration). Next, events were split in their onset and decline phase. The onset phase was considered to last from the 1[st] day until the day of maximum intensity ($I_{max}$), and the decline phase from $I_{max}$-day until the last day of the event. As in Schlegel et al. (2021), for each phase, an ocean mixed layer

heat budget analysis was applied to derive the change in SST attributed to $Q_{net}$, based on the following equation:

$$SST'_{t_2} - SST'_{t_1} = \int_{t_1}^{t_2} \frac{Q'}{\rho_o c_{ph}} dt + R \quad (1)$$

The left-hand side of Eq.1 represents the observed change in SST anomaly ($\Delta SST'_{obs}$) relative to climatology, during a specific

phase. Each phase starts at day $t_1$ and ends at day $t_2$, being the start day and $I_{max}$-day for onset phases, or $I_{max}$-day and end day for decline phases, respectively. On the right-hand side of Eq.1, Q' is the daily anomaly of $Q_{net}$. The latter consists of the latent and sensible heat fluxes and net short-wave and net long-wave radiation (LH, SH, SWR and LWR, respectively), as follows:

$$Q_{net} = LH + SH + SWR + LWR \quad (2)$$


To compute heat flux anomalies, first a daily climatology was computed following the same methodology as with SST for the period 1993-2022. Daily heat flux anomalies were then constructed relative to the mean climatological value over the phase duration (e.g., as in Fewings and Brown, 2019). The time integral of Q' divided by the product of the constant values ρ (seawater density), $C_p$ (specific heat capacity) and h (mixed layer thickness) represents the part of the $\Delta SST'_{obs}$ explained by Q' during this phase ($\Delta SST_Q$).

The second term of the right-hand side in Eq.1 stands for any contribution to $\Delta SST'_{obs}$ from other mechanisms affecting SST: horizontal advection, vertical mixing processes, horizontal diffusion of heat fluxes, radiative heat loss below the mixed layer. The contribution of Q' in driving a MHW onset/decline phase N is then quantified through the following proportion of change:

$$P(N) = \frac{\Delta SST_Q(N)}{\Delta SST'_{obs}(N)} \quad (3)$$




Therefore, a positive heat flux contribution value during a MHW phase indicates a favoring role of heat flux in the corresponding SST evolution, i.e., a warming (cooling) effect of heat flux during onset (decline). Analogously, a negative contribution value during either an onset or a decline phase, indicates that heat flux opposes the corresponding SST tendency.

Finally, to examine the evolution of MLD during MHWs, time-series of cumulative MLD anomalies (MLDA) were constructed for onset/decline phases. These time-series were computed by adding, at each MHW day, the daily anomaly of MLD of the previous day, in order to account for longer time-scales associated with the mixed layer evolution (as in Schlegel et al. (2021)). To explore the correlation between MLDA and SST anomalies (SSTA), Pearson correlation coefficients were computed for onset and decline phases, separately.


**Table 1**

| Product ref. no | Product ID & type | Data Access | Documentation |
|---|---|---|---|
| 1 | SST_MED_SST_L4_REP_OBSERVATIONS_010_021; Satellite observations | EU Copernicus Marine Service Product (2022a) | Product User Manual (PUM): Pisano et al. (2022a) <br><br> QUality Information Document (QUID): Pisano et al. (2022b) |
| 2 | SST_MED_SST_L4_NRT_OBSERVATIONS_010_004; Satellite observations | EU Copernicus Marine Service Product (2022b) | PUM: Pisano et al. (2022c) <br><br> QUID: Pisano et al. (2022d) |
| 3 | MED_MULTIYEAR_PHYS_006_004; Numerical models | EU Copernicus Marine Service Product (2022c) | PUM: Lecci et al. (2022a) <br><br> QUID: Escudier et al. (2022) |
| 4 | MEDSEA_ANALYSISFORECAST_PHY_006_013; Numerical models | EU Copernicus Marine Service Product (2022d) | PUM: Lecci et al. (2022b) <br><br> QUID: Goglio et al. (2022) |
| 5 | ERA5 hourly data on single levels; Numerical models | Copernicus Climate Data Store | Hersbach et al. (2023) <br><br> https://cds.climate.copernicus.eu/cdsapp#!/dataset/reanalysis-era5-single-levels?tab=overview |



## 3. Results

### 3.1 MHW detection

Properties of MHWs exhibit high variability throughout the Mediterranean Sea (Fig. 1a-c). The northwestern part of the basin, along with the northern Adriatic and northern Aegean Seas, experienced on average the highest event intensity over the period 1993-2022, exceeding 2.5 and 2 °C, respectively (Fig. 1b). Events tend to last longer in the eastern part of the basin (Aegean and Levantine Seas), while the shortest durations are mostly found in the Ionian and Alboran Seas (Fig. 1c). The mean event frequency over the study period closely follows the mean intensity spatial distribution (Fig. 1a,b), suggesting that the southernmost flanks of the Mediterranean Sea experience less intense and less frequent MHWs. Results generally agree with literature on MHWs in the Mediterranean Sea during the recent decades (Darmaraki et al., 2019a; Ibrahim et al., 2021; Juza et al., 2022; Dayan et al., 2023), despite some differences in event detection methods (e.g., choice for percentile-based threshold, accounting for MHW spatial extent) or study periods (choice of climatological period, period for event detection).

Increased MHW frequency over the past 30 years is consistently observed across all sub-regions (Fig. 1d), exceeding one event per decade in most of the basin. Also, MHW duration has increased particularly in the eastern basin, with trend values locally exceeding ten days per decade (Fig. 1f). Notably, intensity has not increased over the entire basin during the study period (Fig. 1e). The northernmost regions, which are characterized by higher MHW intensity, present small decreasing MHW intensity trends (though not statistically significant in most cases) which is in agreement with Dayan et al. (2023), and Ibrahim et al. (2021) for the eastern basin.



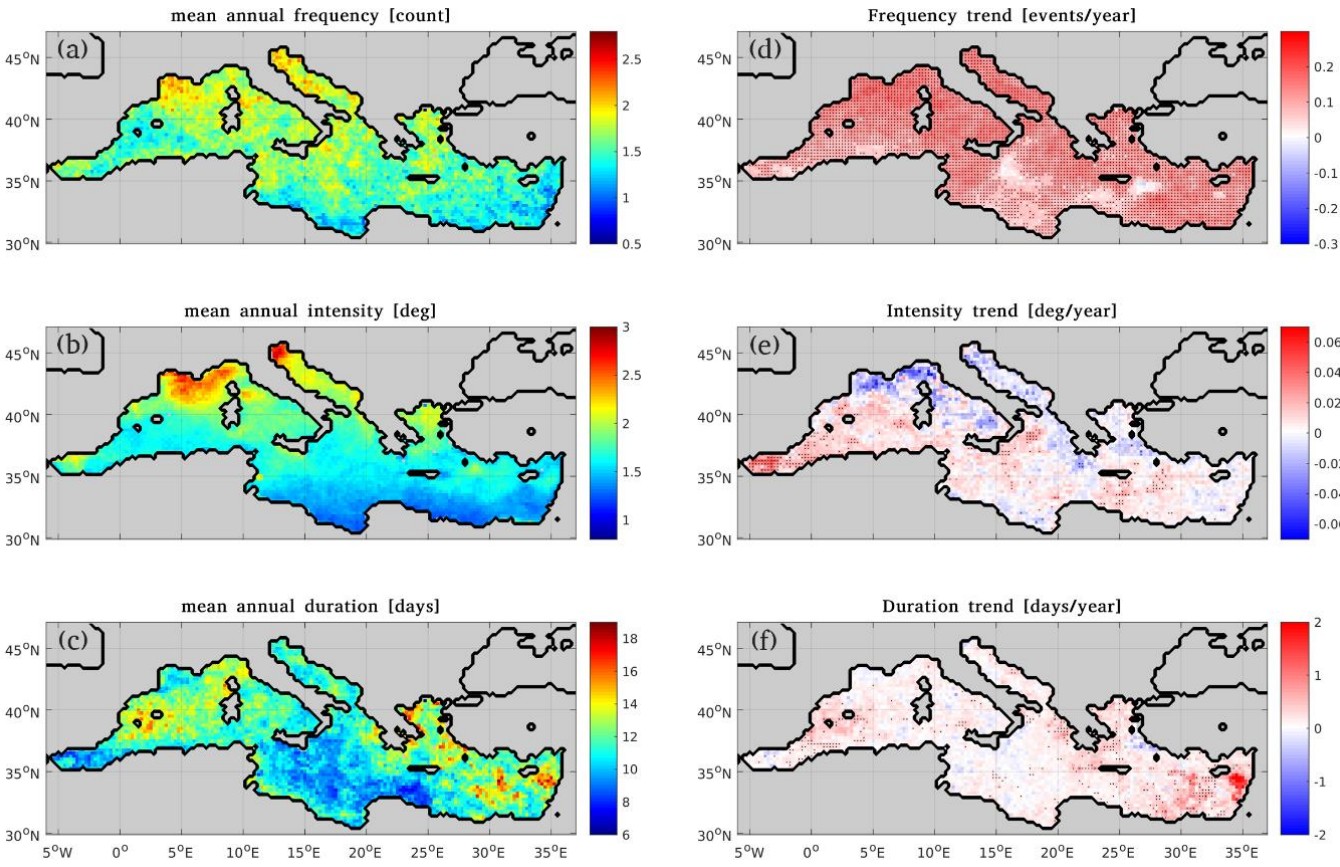

**Figure 1 Left: Mean annual frequency, intensity and duration of MHWs for the period 1993-2022 (a, b, c, respectively); Right: Linear trends of annual values of frequency, intensity and duration (d, e, f, respectively). Black dots superimposed on trend fields correspond to statistically significant trends (Mann Kendall test, 95% confidence level)**

### 3.2 The role of heat flux during MHW onset and decline

Surface heat fluxes contribute to the observed surface warming during the onset phase of the majority of the events (92%) detected in the Mediterranean Sea within 1993-2022 (Fig. 2a). In particular, in 44% of the events, heat flux presents a major role during the development of MHWs, in terms of explaining more than half of the observed change in SST anomaly (Fig. 2b). During the rest of the events (8%), heat fluxes oppose the surface warming over the onset phases; therefore, other mechanisms compensate for their cooling effect and are responsible for the observed warming in such cases.





**Figure 2 a) Contribution of heat fluxes to the observed change in SSTA during MHW onset and decline phases, for the period 1993-2022; b) Percentage of events primarily driven by heat fluxes (i.e., with more than half of the observed warming (cooling) attributed to heat fluxes during onset (decline)) for the entire Mediterranean Sea and sub-regions mapped in (c); d-e) Boxplots for the contribution of heat fluxes during onset (red) and decline (blue) for the entire Mediterranean Sea and the sub-regions; f-g) Boxplots for heat flux contribution per season, for onset (red) and decline (blue); h-i) Boxplots per season, for the contribution of heat flux components during onset (red) and decline (blue); From left to right: Latent heat flux, sensible heat flux, net short-wave radiation, net long-wave radiation. Notes: Positive heat-flux contribution during onset (decline) means that heat-fluxes warm (cool) the sea surface; boxes in boxplots define the interquartile range from the 25th up to the 75th percentile and whisker bars correspond to values falling within 1.5 times the interquartile range.**






Air-sea heat exchange contributes to surface cooling during decline phases in a much smaller percentage of MHWs (58%), while they are the primary driver only in 17% of the decline phases (Fig. 2a,b). In other words, almost half (42%) of the observed MHWs in the basin decay under non-favorable heat flux conditions (i.e., heat flux opposing the SST decrease), while oceanic processes are the dominant driver of most (83%) MHW declines.


Heat flux exchange explains to a great extent the MHW onset in all examined sub-regions (Fig. 2d). The Adriatic and the Aegean Seas stand out with most of the observed heating during onset attributed to heat flux in more than half of the identified events (Fig. 2b). In 6 out of the 7 sub-regions, the percentage of events primarily driven by heat fluxes during their onset ranges between 39%-53%. The Alboran Sea presents a much lower percentage of events attributed to heat fluxes (22%), potentially suggesting an

enhanced role of advective processes in regulating SST in the area, likely associated with its proximity to the Gibraltar Strait. In contrast, the corresponding percentages for the decline phases show less spatial variability across all sub-regions, with only 15-20% of the examined declines found to be mainly driven by heat flux exchange (Fig. 2b).

Seasonal analysis was performed considering winter as the period from December to February, while events spanning different

seasons were assigned to the season when their intensity maximizes. Most of the events were found to occur during summer, followed by spring, autumn and then winter. As expected, the magnitude of SST anomalies during MHWs varies among the seasons (e.g., Thoral et al., 2022), but the contribution of heat flux to their formation is consistently weaker during decline compared to onset phases throughout the year (Fig. 2f,g).

Results suggest a greater contribution of heat flux exchange to the MHW evolution within warmer seasons (from an ocean perspective, i.e., summer and autumn) (Fig. 2f,g). Autumn shows the highest percentage of events driven by $Q_{net}$ in both onset and decline. LH fluxes are mainly regulating the contribution of $Q_{net}$ during both phases and throughout the year, followed by SWR and then SH, while LWR exhibits less clear behavior in all seasons (Fig. 2h,i). With the exception of SH, the relative contribution of each heat flux component presents higher variability during decline compared to onset phases, across all seasons. Specifically,

approximately an equal number of decline cases are associated with positive and negative contributions of the heat flux components to the observed cooling, indicating a less predictable role of $Q_{net}$ during decline compared to onset periods throughout the year. (Fig. 2i). The lowest percentage of heat flux-driven events is observed during the winter onset and spring decline phases. Particularly for spring, heat fluxes during most declines act against the observed surface cooling (mainly through suppressed LH losses) suggesting that MHW dissipation in spring is commonly driven by oceanic factors.


Results suggest that oceanic processes play a key role during onset phases of MHWs, as the air-sea heat fluxes are not the primary driver in slightly more than half (56%) of the examined cases. Yet, surface fluxes are found to contribute to a great extent in the development of MHWs throughout the year, with a dominant role of LH flux. A further weakened role of heat fluxes is found during declines, indicating that MHWs decay is also primarily driven by oceanic processes.



## 3.3 Links with mixed layer depth and MHW characteristics

To gain insight into subsurface conditions during MHWs, we examined how MLD evolves in relation to SST. Negative (positive) correlation between MLDA and SSTA that was found for a large number of events during onset (decline) implies mixed layer shoaling over the entire MHW duration in these cases (Fig. 3a-top). While a MHW event develops, a reduction in MLD is commonly expected, as surface warming may strengthen the stratification of the water column (D'Ortenzio & Prieur, 2012). Given that heat fluxes are found to contribute to the warming phases in most cases, the concurrent mixed layer shoaling found during most onsets may be interpreted as an effect of the warming driven by the atmosphere. Nevertheless, a thinner than usual mixed layer may also exist before the event occurrence and act as a pre-conditioning factor (e.g., Lee et al., 2023). The greatest MLDA-SSTA correlation is found during spring and summer events, while no significant correlation is observed in winter (not shown), as the deeper mixed layer during colder months is expected to be less responsive to surface SST variations (D'Ortenzio & Prieur, 2012). High positive MLDA-SSTA correlation observed during declines implies that the mixed layer in such cases continues to shoal while SST decreases (Fig. 3a), similar to the mixed layer analysis by Schlegel et al. (2021) for the northwest Atlantic. This finding further supports that oceanic processes play a significant role during MHW decline periods, as also indicated by the weaker contribution of heat fluxes during declines. It also suggests that the surface cooling occurring right after the peak intensity day is likely not due to mixing in the vertical (further discussed below).

To gain a better understanding of the relationship between subsurface conditions and the contribution of heat flux exchange at the air-sea interface during the evolution of MHWs, this information is integrated into Fig. 3a. During their onset, MHWs are largely driven by heat flux exchange and most of them are accompanied by mixed layer shoaling (Fig. 3a-top). However, there are events where MLDA are strongly positively correlated with SSTA during onset, indicating surface warming evolves while the mixed layer deepens. Regarding decline periods, almost equal numbers of cases show positive and negative heat flux contributions (Fig. 3a-bottom). While a decrease in MLD is evident in most decline phases (as indicated by correlation coefficient close to one), a significant MLDA-SSTA correlation is absent in a considerable number of decline phases. Additionally, there are cases during declines where a high negative correlation between MLDA and SSTA is observed, indicating mixed layer deepening while SST decreases. Such cases are encountered when heat fluxes contribute to surface cooling (Fig. 3a-bottom), suggesting that vertical mixing (most probably wind-induced), works in the same direction.

MHW onset and decline phases were also examined in relation to their duration. They were grouped in short, medium and long duration, considering phases lasting less than 5 days, from 5 to 10 days and more than 10 days, respectively (Fig. 4a). Shorter onset/decline periods, being the most prevalent category, slightly overshadow the contribution of heat flux during longer lasting MHW phases (Fig. 4a-c vs Fig. 2a). In particular, as compared to medium, followed by long durations, short durations (both for





**Figure 3 a)** Heatmap relating the net heat-flux contribution with Pearson correlation coefficient (CC) between time-series of SST anomalies and time-series of mixed layer depth cumulative anomalies computed for each MHW onset and decline phase (upper and lower heatmap, respectively) for the period 1993-2022. Colors correspond to the number of events falling in each bin. **b)** Boxplots for the distribution of CC values for onset/decline phases lasting less than 5 days, between 5-10 days and more than 10 days (short, medium and long duration phases, respectively); **c)** Percentage change (%) of MLD over a fixed 7-days period after the MHW end day compared to MLD during the onset period. Notes: Positive heat-flux contribution during onset (decline) means that heat-fluxes warm (cool) the sea surface; Negative (positive) CC values close to -1 (+1) during onset (decline) correspond to reduction of the mixed layer depth while SST increases (decreases); Boxes in boxplots define the interquartile range from the 25th up to the 75th percentile and whisker bars correspond to values falling within 1.5 times the interquartile range.

onset and decline) tend to exhibit a smaller contribution of heat flux exchanges triggering the SST evolution (Fig. 4a-c). For long-

duration onset phases, the contribution of heat fluxes surpasses the contribution of oceanic factors, as the former explain more than



half of the surface warming, in more than half of the events (Fig. 4c). Similarly, it is mostly during shorter declines that heat fluxes do not present a systematically positive or negative contribution to the SST decrease, while a shift towards higher contributions of
heat fluxes is found for longer phases (Fig. 4a-c). Importantly, the relationship between MLDA and SSTA becomes less clear during onset/decline phases of medium and, in turn, long duration, as indicated by weaker correlation (Fig. 3b). In the case of decline periods, mixed layer shoaling is more frequently observed during short-lasting declines, suggesting a continuation of mixed layer shoaling already present during the onset in such cases.

These findings suggest that the definition followed for a MHW evolution phase needs to be cautiously taken into account when interpreting results, especially when surface and sub-surface conditions are examined for the same phases. This is due to the time needed for a surface warming signal to penetrate below as well as due to the longer time-scales associated with processes at deeper layers. Notably, at the end day of a surface MHW, being the end day of the decline phase (as defined in this study), SST is still above the event-detection threshold and will fall below this value by the following day. Therefore, events further dissipate after the
decline phase ends. Given such considerations, and depending on the specific purpose of a study, different approaches for defining a decline period may be followed. For instance, Darmaraki (2019c) considered the entire MHW duration as the event development phase and treated the following period as decline, Schlegel et al. (2021) used the same definitions as the present study, while Marin et al. (2022) considered a dynamic criterion for defining the decay period, including a larger portion of the temperature change during the event dissipation after the MHW end day.


On these grounds, we additionally examine how MLD evolves after the MHW end day, to shed light on what follows the progressively increasing stratification we observed during MHWs. To this aim, we compute for each event the percentage change in the mean MLD between the onset period and a fixed 7 days-period following the MHW end day (Fig. 3c). Results reveal the existence of a deeper mixed layer after the decline of most events (83.5%) compared to the onset period. In these cases, the
magnitude of change in MLD is also significantly greater, occasionally exceeding 100%, compared to instances when the mixed layer is thinner during this period than during the onset (changing the predetermined length of the examined post-decline period does not significantly alter these conclusions). This analysis shows that a significant mixed layer deepening most likely occurs after the end of MHW decline phases in the Mediterranean Sea, suggesting that vertical mixing eventually contributes to the heat dissipation.


Considering the continuation of mixed layer shoaling found mostly during shorter declines, results suggest that the oceanic factor regulating the SST decrease after the peak intensity day of most events is probably heat advection, while vertical mixing in these cases becomes important over the following days. In line with our findings, Marin et al. (2022) found that the principal driver of the upper ocean temperature changes during MHW onset and decline phases in mid latitudes is horizontal heat advection. However,
their results cannot be directly inter-compared to ours, as they use modelled depth-integrated upper ocean temperatures (thus not detecting surface events), they focus only on extreme cases and they employ a different definition of onset/decay period.





Finally, results were examined in relation to MHW intensity and severity, to account for the perspective of MHW extremity in terms of absolute and normalized SST anomalies, respectively. Events were split based on their severity category following the

categorization scheme proposed by Hobday et al., 2018 (definitions included therein). Results suggest a smaller contribution of heat fluxes in the evolution of more severe events (Fig. 4d-f). Half of the moderate events were found to be mainly driven by heat fluxes, while this percentage decreases for strong, severe and extreme events, as illustrated in Fig. 4d-f. This may be associated with the enhanced role of heat fluxes during longer events, as was previously discussed, and the fact that events of higher severity categories tend to present shorter duration. Similar results were found when examining the contribution of air-sea heat fluxes in relation to the

mean intensity of MHWs (not included). Nevertheless, further investigations are needed to unravel how our methodological choices affect these findings.

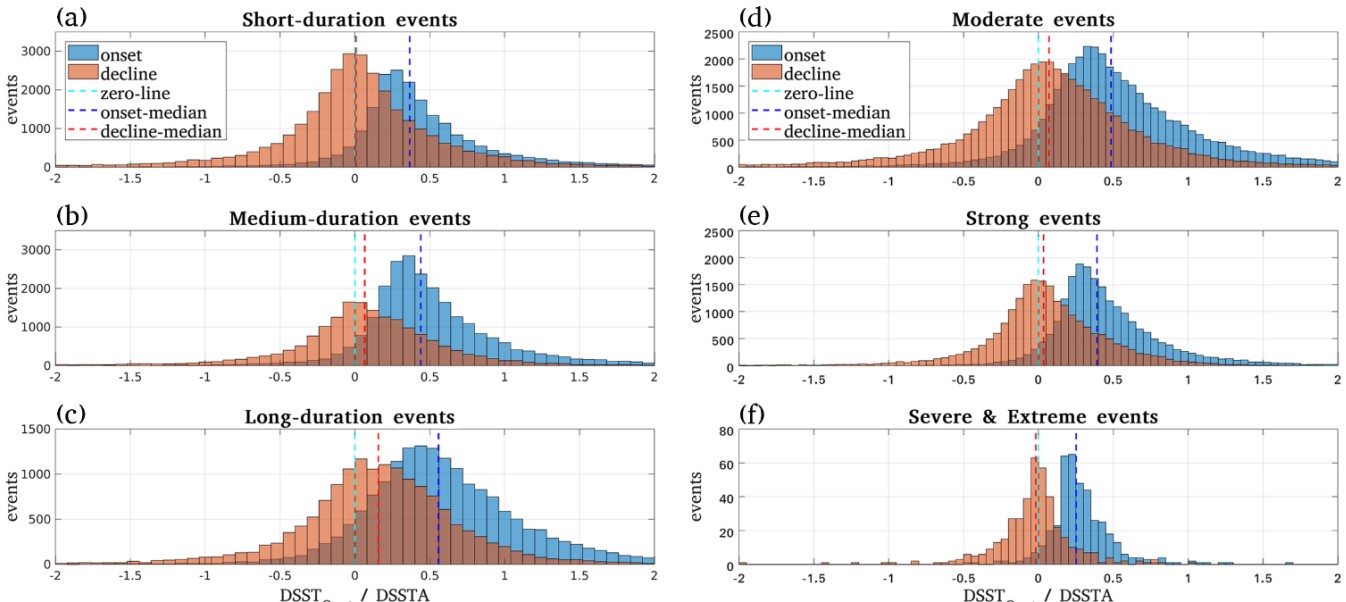

**Figure 4 Left: Contribution of heat fluxes to the observed change in SSTA during MHW onset and decline phases, for the period 1993-2022, for phases lasting: less than 5 days (a), between 5-10 days (b) and more than 10 days (c) (short, medium and long duration phases, respectively). Right: Same as left but for different MHW severity categories: moderate (d), strong (e) and severe/extreme (f), based on the categorization scheme by Hobday et al. (2018).**

## 4. Conclusions

This section investigates the relative role of air-sea heat exchange and oceanic processes during MHWs in the Mediterranean Sea. Events were initially identified based on satellite-derived SST within 1993-2022. Then, an ocean mixed layer heat budget analysis

was performed to derive the change in SST attributed to the net surface heat budget during onset and decline phases.





Air-sea heat fluxes are the primary driver in 44% of the onset and only in 17% of the decline phases in the basin, suggesting that oceanic processes have a key role in regulating SST during MHWs, particularly during decline periods. Heat fluxes act in favor of the development of most MHWs, across all seasons and especially during warmer months and onset phases. Moreover, their
contribution is greater in the Adriatic and Aegean sub-basins, where they become the major driver of most onset phases. Among the heat flux components, LH emerges as the most significant contributor to SST anomalies, in line with prior studies (Sen Gupta et al., 2020; Oliver et al. 2021; Schlegel et al., 2021; Marin et al., 2022). Short-lasting onset/decline phases (shorter than 5 days) tend to experience a smaller contribution of heat flux in forming the SST evolution, compared to longer phases (lasting 5-10 or more than 10 days). Furthermore, there is an inverse relationship between MHW severity and the contribution of heat fluxes.


Examining the mixed layer during MHWs revealed a progressively decreasing MLD over the entire event-duration, particularly for shorter-lasting events. In turn, a significant mixed layer deepening was found to occur after the end of the decline period of most events in the basin. In cases of smaller contribution of heat fluxes (e.g. during shorter compared to longer events, or during declines compared to onsets), a stronger correlation between SSTA and MLDA is also found, further supporting the key-role of oceanic
processes in such cases. These findings suggest that the surface cooling occurring right after the peak intensity day is likely not associated with vertical mixing. Moreover, this potentially suggests that horizontal advection is the oceanic factor playing the most significant role by the time the decline period begins, especially for shorter declines. This hypothesis finds support in the results of Marin et al. (2022), who highlighted the role of horizontal heat advection during the MHW evolution, and Schlegel et al. (2021) who suggested that advection and mixing should drive the MHW decline, based on similar indications.


Nevertheless, authors suggest taking into account potential limitations associated with the definition of MHW phases followed within a study, especially while examining concurrent sub-surface conditions. These considerations concern the longer time-scales below the sea surface, as well as the complexity associated with long-lasting events which are not expected to be adequately described by two single phases. Specifically, subsequent warming and cooling periods may occur within a long onset or decline
phase, complicating their representation by the definition employed in this study for onset and decline. Despite these caveats, this study provides useful insights into the role of surface heat fluxes and mixed layer dynamics during MHWs in the Mediterranean Sea. Importantly, following the methodology applied by Schlegel et al. (2021) allowed for a fair inter-comparison of results among two substantially different regions (Mediterranean Sea vs northwest Atlantic). Results are surprisingly similar, which is largely attributed to the same methodology employed to assess drivers of SST-defined events, and especially to the same definition followed
for onset and decline periods. The striking similarities in the results for the two regions may also imply inherent characteristics of the interplay of air-sea interaction and oceanic processes during anomalous SST fluctuations over similar latitudinal zones, which is observed to a certain extent also in Marin et al. (2022).

This study highlights the need for considering subsurface information in MHW studies, to gain insight into ocean internal dynamics
throughout their evolution. In this context, combining observations and ocean reanalysis systems, such as in Dayan et al. (2023),

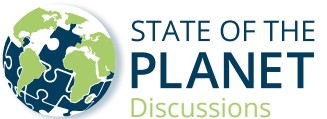

and considering MHW evolution periods aligned with the objectives and specific characteristics of a study appears to be a promising direction towards understanding physical drivers, improving monitoring, and therefore enabling early warning of MHWs.

## Data availability

Information on the products used in this paper is included in Table 1.

## Author contribution

DD defined the research problem. DD conducted the analysis and wrote the manuscript, with contributions from GK and MH. All authors contributed to the interpretation of results.

## Competing interests

The contact author has declared that none of the authors has any competing interests.

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
