# Peer review of "The role of air-sea heat flux for marine heatwaves in the Mediterranean Sea"

_State of the Planet, 2023_

## Referee Comment (RC2)

**General Comments**

The authors examine the role of air-sea heat flux during Marine Heatwaves (MHWs) events in the Mediterranean Sea over the last 30 years. These events were identified using satellite-derived Sea Surface Temperature (SST) data from 1993 to 2022. An analysis of the ocean mixed layer heat budget was conducted to determine the change in SST attributed to the net surface heat budget during onset and decline phases of MHWs. Air-sea heat fluxes are found to be the primary driver of most MHW onsets, particularly in warmer months and during onset phases, while oceanic processes play a key role in regulating SST during decline periods. A progressively decreasing mixed layer depth (MLD) is observed over the entire event duration, particularly for shorter-lasting events, with significant mixed layer deepening occurring after the end of the decline period. This study underscores the importance of considering subsurface information to better describe the evolution of these extreme events. Combining observations and ocean reanalysis systems appears promising for improving monitoring and early warning of MHWs.

In general, this paper is well-organized and presented in a coherent manner. While the findings may not be groundbreaking, they are relevant within the context of the Mediterranean Sea and can contribute to the advancement of knowledge on this topic.

My main concern, however, is the emphasis placed by the authors on oceanic processes, such as horizontal advection and vertical mixing. Specifically, some findings are just deduced by the authors without conducting a thorough analysis of these processes (e.g., lines 12-13; 227-229; 231-233; 265-268 and so on). Hence, I suggest either revising the sentences highlighting oceanic processes or providing additional analysis to support the findings.

**Specific Comments**

[Lines 12-13]. "Our findings suggest that oceanic processes...". Based on my last comment given in the General Comments, I would rephrase this sentence giving more emphasis to the role of heat fluxes, which is the topic of this work.

[Line 44]. I suggest the authors the following reference that investigates the role of atmospheric forcing and wind-driven mixing during the 2022/2023 MHW event in the Mediterranean Sea.

Marullo, S., Serva, F., Iacono, R., Napolitano, E., di Sarra, A., Meloni, D., ... & Santoleri, R. (2023). Record-breaking persistence of the 2022/23 marine heatwave in the Mediterranean Sea. Environmental Research Letters, 18(11), 114041. https://doi.org/10.1088/1748-9326/ad02ae

[Line 59]. Please, expand acronyms: NRT CMEMS.

[Line 68-69]. Please, clarify how the climatology was computed (is it just an average or did you apply a smoothing window?)

[Table 1]. For your information, here are the references for products n.1 and n.2:

(a) Product n.1. Pisano, A., B. Buongiorno Nardelli, C. Tronconi, and R. Santoleri (2016). The new Mediterranean optimally interpolated pathfinder AVHRR SST Dataset (1982 – 2012). Remote Sensing of Environment, Vol. 176, pg. 107-116. http://dx.doi.org/10.1016/j.rse.2016.01.019

    (b) Product n.2. Buongiorno Nardelli, B., Tronconi, C., Pisano, A., and Santoleri R. (2013). High and Ultra-High resolution processing of satellite Sea Surface Temperature data over Southern European Seas in the framework of MyOcean project. Remote Sensing of Environment, Vol. 129, pg. 1-16. http://dx.doi.org/10.1016/j.rse.2012.10.012

[Line 113]. "Events tend to last longer in eastern part...". I recommend to include the central-western region of the Mediterranean into this consideration as well.

[Line 114-115]. "...frequency...closely follows intensity...". Honestly, I do not see this 'high correlation'. I recommend to quantify the correlation or rephrase the sentence.

[Line 121-125]. I recommend to quantify the trends of intensity, duration and frequency with confidence intervals as well.

[Figure 1]. Concerning the trend maps (d-e-f), I would suggest to put black dots over non-significant pixel values (that is, just switch the overlapping criterion).

[Figure 2]. The label for x-axis is DSSTQnet/DSSTA while you used $\Delta SSTQnet/\Delta SSTA$ in eq. 3. I recommend to adopt the same notation. Same comment for Figures 3 and 4.

[Line 166-167]. This sentence is somewhat misleading and complex ("are not the primary driver in..."). It appears to contradict what is stated at line 129. I would suggest rephrasing it. Overall, I recommend greater clarity when distinguishing between the roles of heat fluxes and oceanic processes, as in some cases one is more significant than the other and, in other cases, the opposite.

[Line 186-188]. This sentence is a repetition of what already stated above. I suggest to rephrase or remove it.

[Line 192]. "a significant MLDA-SSTA correlation is absent...". What do you mean with this sentence? To me is not clear.

[Line 248]. "...and oceanic processes". I would suggest to substitute oceanic processes with mixed layer heat budget analysis (or something equivalent).

---

## Author Response (AR1)

**General comment by authors:**

We would like to thank both reviewers for the effort and care with which they assessed our submission. We have revised the manuscript following their excellent recommendations. Please, find below a point-by-point response to their comments, followed by a separate section listing the changes implemented in the manuscript. For ease of reference, the reviewer's comments are presented in blue font, while the authors' responses are presented in black font.

**Reply to Reviewer 1:**

**Summary**

The authors use a bevy of remotely sensed, reanalysis, and modelled products to determine the interplay between SST, Qnet, and MLD in the Mediterranean Sea from 1993-2022. More specifically, the authors investigate what the physical drivers are of MHWs during the onset and decline portions of the these events by decomposing the contribution of specific components of the mixed-layer heat budget equation o the anomalous SST observed during a MHW. These results are also parsed out by season, region, length of the event, and category (e.g. Moderate, Strong, etc.). The methodology used here is previously published, but has not yet been applied to the Med. The results presented here are an important contribution to the MHW literature and I think this paper is going to be very well cited over the coming years.

I have no substantive comments on the intro, methods (which is rare), or conclusions. I only have two minor issues (repeated in the specific comments below). The first is that the authors appear to not have accounted for the loss of shortwave radiation out of the bottom of the MLD in their mixed-layer heat budget equation. This will likely have little impact on the results, but it should either be accounted for, or the authors should argue for why they aren't doing it. The second is that while I found the authors criticism of the methodology for how to account for the onset/decline periods, they unfortunately stop short of providing any sort of advice on what could be done instead.

All the best,

-anonymous

**Title**

- Consider "heat flux" rather than "heat fluxes". Heat flux is generally used as a non-count plural noun. As the authors prefer.

Thank you for your comment. We have incorporated this suggestion in the revised manuscript.

**Abstract**

- ln 9: "research studies" is a tautology. Rather choose one word or the other.

Thank you, this has been corrected.

- ln 10: "Marine Heatwaves" -> "Marine heatwaves" Or as the editor prefers.

Thank you for this suggestion, we have included it in the revised manuscript (leaving the final decision to the editor).

- ln 16: "Moreover..." Which direction is the relationship? Is their more or less input of heat flux with increased MHW severity?

Results shown in Fig. 4 suggest a smaller contribution of heat flux for events of higher severity, and vice versa. For this reason, we report an "inverse" relationship between the two quantities in the sentence: "*Moreover, an inverse relationship between MHW severity and the contribution of heat fluxes is observed*", i.e., the greater the severity, the lesser the contribution of heat flux.

 ln 21: "have key" -> "have a key"

Thank you, this has been corrected.

- ln 23: "emphasizes the need" Almost all studies of surface MHWs end by saying there is a need for the consideration of subsurface information. The criticism of the limitations caused by the onset/decline phase definition is interesting.

Thank you for your comment. You are right in noting that most recent studies focus or suggest focusing on sub-surface events, which is mainly due to higher relevance to marine life. While this work treats only SST-based events, examining the mixed layer evolution (being the sub-surface information considered here) provided useful insights relevant to the surface events. It is taking this into account that we included this note on the importance of sub-surface information in our concluding remarks.

**Introduction**

- ln 49: "Darmaraki et al. (2023)" Hmm... I'm not sure a talk at a conference can be referenced here. This is not peer reviewed work. I don't doubt the accuracy of the work though. The authors may want to edit the sentence slightly to account for the non-peer reviewed nature of these findings.

Thank you for your suggestion, we have removed this reference.

**Data and Methods**

- ln 58: It would be nice to list the horizontal resolution of the other products in this paragraph as well.

Thank you for your suggestion, we have added this info in this paragraph.

- ln 61: "ref. no. 05" Why not list the products in the text and the table in the same order?

Thank you for noting this, the order has been corrected.

- ln 85: Did you account for the amount of shortwave radiation that passes out of the upper mixed layer?

We thank the reviewer for this insightful comment. Indeed, we have used a simplified approach that does not account for the penetration of solar radiation below the mixed layer. A relevant note has been added for clarity in the revised manuscript (Methods).

Considering the Jerlov Water Type IA for relatively clear sea water, 77% of the solar radiation is expected to be absorbed within the upper 10 meters of the ocean (as computed based on the solar radiation attenuation equation in Paulson and Simpson,1977).  On these grounds, and taking into account the minimum mixed layer depth values over the study period from the utilized reanalysis dataset (deeper than 11m as presented in the figure below), we have assumed that the followed approach does not significantly affect our conclusions.

[Figure]

Minimum mixed layer depth values during the study period derived from the utilized Med-Physics reanalysis dataset

To verify this, preliminary results have been produced incorporating the discussed parameterization (presented in the boxplots below). These results show the expected effect of the parameterized heat loss, i.e., a slight suppression (enhancement) of the contribution of air-sea heat flux in driving the SST onset (decline). Very small differences are observed and results are in line with the existing ones concluding on the dominant role of oceanic processes during both phases.

We should note here that this study covers all MHW cases in the basin over the past 30 years. Therefore, a number of approximations have been considered. For example, the definition for onset and decline periods is expected to introduce uncertainties especially during long-lasting ones, as discussed in the manuscript. In addition, the heat budget analysis performed in this study represents a first step towards investigating the role of the air-sea heat flux in the evolution of MHWs in the

basin from a statistical perspective. This role is assessed with respect to a residual term (i.e., the non-heat flux terms are combined into a single one), which represents the cumulative effect of all other factors influencing the SST tendency during a MHW phase. We therefore believe that our findings on the relative role of these two terms are sufficiently robust within the context of the level of detail considered and the uncertainties associated with this study.

[Figure]

Box plots for the contribution of heat flux during onset (red) and decline (blue) of Mediterranean MHWs during 1993-2022. Left: current results without considering the penetration of solar radiation below the mixed layer (EXP0), Right: including this parameterization (EXP1).

**Results**

- ln 121: Reporting on MHW frequency is a bit of a tricky thing. This is because, particularly in the Med, we are beginning to approach the time when we have fewer MHWs as they start to last long enough to merge into one long MHW per year. I therefore always recommend to report on total MHW days per year, rather than frequency.

Thank you very much for this suggestion. We totally acknowledge this concern and we can certainly follow the suggested approach in future work, as MHW days are free of this limitation. Nonetheless, we find a statistically significant positive trend for the count of events per year almost over the entire basin (Fig. 1d in the manuscript). The very few exceptions (e.g., to the southeast of Sicily) do not present negative frequency trends, they simply lack statistically significant frequency trends. Therefore, we believe a clear message regarding the increasing count of events per year can confidently be derived from the analysis of this dataset for the considered period.

- ln 136: This sentence is confusing. After reading it a couple of times I understand what the authors are trying to say. It is unfortunately the nature of trying to explain this concept using words. It's not easy to do and I don't have a better recommendation.

- ln 249: "definition..." I agree with the authors to not give the full category definition here.

- ln 243: "events of higher severity categories..." I'm not convinced this is factually accurate.

We thank the reviewer for bringing their attention to this. First, let it be noted that the index we use to address the MHW severity here is the following:

$$S_{i,j,t} = \frac{T_{i,j,t} - T_{i,j,d}^{clim}}{T_{i,j,d}^{90th} - T_{i,j,d}^{clim}}$$

where T is the maximum SST during the event at the i, j location at day t, Tclim is the mean climatological SST at the i, j location for the climatological day-d (that corresponds to the actual day-t), and T90th is the threshold calculated for that location and climatological day. This severity index does not account for the event duration. It represents the event severity in terms of extreme temperatures with respect to local climatological variability. The relationship between the role of heat flux and severity could potentially differ if an alternative index, such as the cumulative intensity during each event, had been employed.

In the figure below we map the used severity index (S) against the duration of onset and decline phases (top and bottom panel, respectively), alongside the count of events. This figure shows that the *severe* and *extreme* cases (corresponding to S>3 according to the categorization scheme of Hobday et al, 2018) actually exhibit shorter durations. We do not claim that the smaller heat flux contribution observed for events of higher S (Fig. 4d-f in the manuscript) arises from the relationship between S and duration as observed in this preliminary test. However, results from this test are potentially associated with our findings for the heat flux contribution during events of different S and were thus considered worth noting. Importantly, as noted in the manuscript, further investigations are needed to unravel how our methodological choices affect these findings.

[Figure]

*Figure: Distribution of MHWs in relation to their severity based on the continuous severity index (S) and the duration of the onset and decline phase (top and bottom panel respectively). S ranges (1-2], (2,3], (3,4] and higher than 4 correspond to Moderate, Strong, Severe and Extreme categories respectively. Note: the colorbar is adjusted to allow for visualizing bins with fewer events (a much higher upper limit corresponds to the actual count of events within 1993-2022 throughout the basin).*

**Conclusions**

- ln 248: "This..." Delete or merge this sentence with the next one. Not necessary to make this statement.

Thank you for this comment. These sentences have been merged as follows: "*This study investigates the relative role of air-sea heat exchange during MHWs in the Mediterranean Sea, using satellite and reanalysis data within 1993-2022.*"

- ln 265: "These findings..." This is interesting, and makes me think of the basin-wide 2022 event. If oceanic advection is one of the primary drivers of the decline of events, but the whole basin is anomalously warm, and no wind exists to cause vertical mixing, then that would explain how a basin scale event can persist for months. That could be important for a future Med that will be much warmer than now.

Thank you for sharing this thought. Indeed, we believe that this mechanism is most likely responsible for a large part of the spatial and temporal extent of such events. However, the role of oceanic advection in this work is inferred from our analysis and has not been directly investigated. For this reason, we briefly discuss this factor solely in terms of its potential role right after the maximum-intensity day, as implied from our actually quantified results (i.e., the contribution of surface heat flux), and the enhanced/suppressed vertical mixing as indicated from mixed layer deepening/shoaling.

- ln 727: "These..." I agree. But what do we do instead? It would be nice if the researchers could propose one or two ideas based on their experiences.

Thank you for this comment. You are right in noting that our suggestion is limited to considering MHW evolution periods aligned with the objectives and specific characteristics of a study. We share this concern and believe that it lies within the wider discussion on challenges associated with the lack of a standardized framework for analyzing MHWs and their drivers in particular. While methodological choices such as the selection of temperature thresholds or the reference climatology (stable or moving) are often supported in relation to impacts on marine species, a series of different approaches have recently been used in quantifying the contribution of physical drivers in absence of a relative discussion. A comparison of methods for quantifying the contribution of potential driving factors has not been performed yet. However, we believe that diverse methodological approaches, such as the definition for MHW phases, can influence findings on MHW drivers. For instance, the continuation of the mixed layer shoaling observed during most decline periods in our study initially surprised us; these results prompted us to examine post-decline periods, revealing delayed (in relation to the considered MHW end day) deepening of the mixed layer. This task highlighted the influence of the methods used to analyze surface vs subsurface information. Similarly, the use of different integration depths when studying drivers is expected to yield varying findings. Therefore, alongside recommending the use of definitions and methods aligned with the specific contexts of individual studies (e.g. SST-based events or events detected based on integrated depths), we underscore the importance of accounting for the associated limitations when interpreting and discussing results. Although we do not propose specific methodological choices, we aim to highlight that clearly articulating the employed ones within a study is vital both for precise interpretation of the corresponding results and for meaningful comparisons across different studies on MHW drivers. This has been stated in a clearer way in the revised manuscript.

**Table 1**

I think it would be useful to give columns for the horizontal resolution, time step, and time series start to end date.

Thank you for this suggestion. We agree, however the content of this Table follows the specific guidelines provided in the context of the Ocean State Report. To our understanding, these

guidelines aim to reduce the datasets-related information within the papers to be included in this special issue, while ensuring that such info can be easily accessed through the references to documentation provided in a table of identical format for all papers. Following your previous comment on products' information in Section Data and Methods, we have enriched this section (but not the Table) adding the spatial resolution of the datasets.

**Figure 1**

**Change labels** for "[deg]" to "[degC]" or somehow indicate it is degrees Celsius. Same for **"[deg/year]".** If the **longitude labels** are only used for the bottom panels, then the **latitude labels** only need to be used for the left-hand panels. As the authors prefer. Why is a Mann Kendall test used and not a simple linear regression? Are the data significantly non-normally distributed? Generally speaking it is no longer preferable to use the rainbow **colour palette**. Rather used one of the viridis colour palettes instead (e.g. https://www.mathworks.com/matlabcentral/fileexchange/51986-perceptually-uniform-colormaps).

I like how the Atlantic coastline provides a natural little border for the panel labels :)

Thank you for your suggestions. Units and coordinates in labels have been updated accordingly and the colour palette has been changed as suggested. As regards the computation of trends, linear regression has been used to estimate the trend values. The Mann Kendall test has been applied to assess the statistical significance of the observed trends. The use of linear regression allows us to quantify the magnitude and direction of trends while the Mann Kendall test, being a non-parametric method, allows for the assessment of statistical significance without relying on the assumption of normality.

Please, also note that we have additionally switched the overlapping criterion for statistical significance putting black dots over the non-significant pixel values as suggested by the 2nd reviewer.

**Figure 2**

Panel a, colours should be switched for onset/decline to match the rest of the panels. Panel b, I think the comparison of onset vs decline will be communicated better if the bar plots are next to each other, rather than being stacked. Panels d-i, consider filling in, rather than line colouring, for the boxplots. It will make them look more substantial. I agree with the authors choice to limit the y-axis range for the boxplots to +-2 values, allowing longer tales to stretch outside of the plotting range.

Thank you for your comments and suggestions. Figure 2a has been updated with red and blue colors used for onset and decline respectively (the same for Fig. 4). Figure 2c has also been replaced to correct a typo in the latitude label. We kept however the line-colouring for boxplots instead of filling them in, as this allowed for a clearer visualization of the median and zero lines in the cases of thinner boxes.

**Figure 3**

Same consideration for boxplots as figure 2. Otherwise I like the layout of the panels. I am wondering if there are any spatial patterns across the Med that may explain some of the onset during MLD deepening MHWs, and vice versa?

Thank you for your comments. Analysis of spatial differences of these specific findings across the basin has not been implemented in the context of this paper. However, we are planning to proceed with a more detailed analysis to address this question also combining other parameters (such as wind speed in relation to the observed mixed layer evolution). Such an analysis is expected to allow for a deeper interpretation of the findings presented in Fig. 3 across the basin, and therefore an improved understanding of physical mechanisms during onset/decline at sub-regional scale.

**Figure 4**

Swap colours for onset and decline to match other figures. This is an interesting way of visualising these complex results.

Thank you for your suggestion. Figure 4 has been updated accordingly.

**Reply to Reviewer 2:**

**General Comments**

The authors examine the role of air-sea heat flux during Marine Heatwaves (MHWs) events in the Mediterranean Sea over the last 30 years. These events were identified using satellite-derived Sea Surface Temperature (SST) data from 1993 to 2022. An analysis of the ocean mixed layer heat budget was conducted to determine the change in SST attributed to the net surface heat budget during onset and decline phases of MHWs. Air-sea heat fluxes are found to be the primary driver of most MHW onsets, particularly in warmer months and during onset phases, while oceanic processes play a key role in regulating SST during decline periods. A progressively decreasing mixed layer depth (MLD) is observed over the entire event duration, particularly for shorter-lasting events, with significant mixed layer deepening occurring after the end of the decline period. This study underscores the importance of considering subsurface information to better describe the evolution of these extreme events. Combining observations and ocean reanalysis systems appears promising for improving monitoring and early warning of MHWs.

In general, this paper is well-organized and presented in a coherent manner. While the findings may not be groundbreaking, they are relevant within the context of the Mediterranean Sea and can contribute to the advancement of knowledge on this topic.

My main concern, however, is the emphasis placed by the authors on oceanic processes, such as horizontal advection and vertical mixing. Specifically, some findings are just deduced by the authors without conducting a thorough analysis of these processes (e.g., lines 12-13; 227-229; 231-

233; 265 268 and so on). Hence, I suggest either revising the sentences highlighting oceanic processes or providing additional analysis to support the findings.

We thank the reviewer for this comment. Indeed, the heat budget analysis performed in this work quantifies only the role of air-sea heat flux in the evolution of MHWs in the Mediterranean basin. This role is assessed in relation to a single residual term (i.e., the non-heat flux terms merged into a single one), representing the cumulative effect of all other (oceanic) factors influencing the SST tendency during a MHW phase. Therefore, the role of oceanic processes is deduced from this analysis and is not directly investigated. However, we additionally examine the evolution of MLD, considering that the progressive increase (decrease) we find for MLD during different MHW phases suggests the enhancement (suppression) of vertical mixing. Such findings are qualitative but we believe they constitute important complementary information.

Sentences highlighting the relevant role of oceanic processes have been revised in the updated manuscript putting more focus on the actually quantified contributions. A clarification has also been added in the Methods section based on the above. Our specific answers (and comments on the example lines mentioned in "General Comments" by the reviewer) are provided below.

**Specific Comments**

[Lines 12-13]. "Our findings suggest that oceanic processes…". Based on my last comment given in the General Comments, I would rephrase this sentence giving more emphasis to the role of heat fluxes, which is the topic of this work.

Thank you for your comment. We agree on re-orienting the presentation of these results. We have revised this sentence (Lines 12-13) as follows: *Results show that air-sea heat flux is the major driver in 44% of the onset and only 17% of the decline MHW phases. Thus, these findings suggest that oceanic processes play a key role in driving SST anomalies during MHWs, particularly during declines.*

[Line 44]. I suggest the authors the following reference that investigates the role of atmospheric forcing and wind-driven mixing during the 2022/2023 MHW event in the Mediterranean Sea.

Marullo, S., Serva, F., Iacono, R., Napolitano, E., di Sarra, A., Meloni, D., ... & Santoleri, R. (2023). Record-breaking persistence of the 2022/23 marine heatwave in the Mediterranean Sea. Environmental Research Letters, 18(11), 114041. https://doi.org/10.1088/1748-9326/ad02ae

Thank you for this suggestion. This paper has been included in the reference list.

[Line 59]. Please, expand acronyms: NRT CMEMS.

Thank you for your comment, acronyms have been expanded.

[Line 68-69]. Please, clarify how the climatology was computed (is it just an average or did you apply a smoothing window?)

Thank you for your comment. To compute the climatology at each grid point, a time-window of 11 days was employed, centered on the day when each daily climatological value was computed. Additionally, a 30 day-window was applied for smoothing the daily threshold time series.

We added this information in the revised manuscript.

[Table 1]. For your information, here are the references for products n.1 and n.2:

(a) Product n.1. Pisano, A., B. Buongiorno Nardelli, C. Tronconi, and R. Santoleri (2016). The new Mediterranean optimally interpolated pathfinder AVHRR SST Dataset (1982 – 2012). Remote Sensing of Environment, Vol. 176, pg. 107-116. http://dx.doi.org/10.1016/j.rse.2016.01.019

(b) Product n.2. Buongiorno Nardelli, B., Tronconi, C., Pisano, A., and Santoleri R. (2013). High and Ultra-High resolution processing of satellite Sea Surface Temperature data over Southern European Seas in the framework of MyOcean project. Remote Sensing of Environment, Vol. 129, pg. 1-16. http://dx.doi.org/10.1016/j.rse.2012.10.012

Thank you for sharing these references. However, the reference format currently included in the manuscript for the Copernicus Marine products follows the specific guidelines provided in the context of the Ocean State Report.

[Line 113]. "Events tend to last longer in eastern part…". I recommend to include the central-western region of the Mediterranean into this consideration as well.

Thank you for noting this, it has been added in the revised manuscript.

[Line 114-115]. "…frequency…closely follows intensity…". Honestly, I do not see this 'high correlation'. I recommend to quantify the correlation or rephrase the sentence.

We thank the reviewer for this comment. Figure 1a shows that the northwestern Mediterranean Sea (around the Gulf of Lions, and to the east of Corsica) and the northern Adriatic Sea, followed by the Aegean Sea, present the higher MHW frequency. These are the areas where MHW intensity also shows its higher values across the basin (Fig. 1b). In addition, the lowest intensity values are observed across the African coasts (10$^o$E eastwards) and it is in several spots within this extended area that the lowest frequency values are also encountered. We agree that "closely follows" written in the manuscript implies a strong spatial correlation among these quantities which is not correct and should be rephrased. To report the observed similarities in the spatial patterns of the discussed fields without overstating their correspondence, we have revised the sentence as follows:

*The mean event frequency over the study period shows some similarities with the mean intensity spatial distribution, suggesting that the most (least) intense and most (least) frequent MHWs are encountered in the northernmost (southernmost) flanks of the Mediterranean Sea (Fig. 1a,b).*

[Line 121-125]. I recommend to quantify the trends of intensity, duration and frequency with confidence intervals as well.

Thank you for this suggestion. Trend values for the entire basin and their 95% confidence interval have been briefly reported in the revised manuscript in addition to the existing discussion on trends in this paragraph. Trends for frequency, mean intensity and duration for the basin are $0.1 \pm 0.06$ events/year, $0.008 \pm 0.02$ degC/year (non-significant), and $0.17 \pm 0.15$ days/year, respectively.

They are computed based on annual values for the entire basin as shown in the draft figure below. No additional results or discussion on long-term trends has been included in the manuscript (out of the focus of this study).

[Figure]

[Figure 1]. Concerning the trend maps (d-e-f), I would suggest to put black dots over non significant pixel values (that is, just switch the overlapping criterion).

Thank you for this suggestion. We have incorporated it in the updated Fig. 1 in the revised manuscript.

[Figure 2]. The label for x-axis is DSSTQnet/DSSTA while you used ΔSSTQnet/ΔSSTA in eq. 3. I recommend to adopt the same notation. Same comment for Figures 3 and 4.

Thank you for noting this, this has been corrected in the revised manuscript.

[Line 166-167]. This sentence is somewhat misleading and complex ("are not the primary driver in..."). It appears to contradict what is stated at line 129. I would suggest rephrasing it. Overall, I recommend greater clarity when distinguishing between the roles of heat fluxes and oceanic processes, as in some cases one is more significant than the other and, in other cases, the opposite.

We thank the reviewer for this comment. We agree that this sentence should be rephrased for clarity. Line 129 states that there is a positive contribution of surface heat flux in 92% of the onset phases. As explained in Methods (Lines 96-98), a positive heat flux contribution during a MHW phase means that the heat flux promotes the observed change in SST anomaly during that phase (e.g., $\Delta SST_Q > 0$ & $\Delta SST'_{obs} > 0$ for onset). However, this does not suggest that heat flux is the primary driver during that phase (in terms of explaining at least half of the observed change in SST anomaly), so there is no actual contradiction with Lines 166-67.

As stated in lines 166-176, the air-sea heat flux is found to be the primary driver of 44% of the onset phases. In other words, heat flux is not the primary driver in 56% of the onset phases. This suggests that in 56% of the onsets, oceanic processes have a dominant role. To increase clarity and avoid misleading the reader, we have rephrased Lines 166-69, as follows:

*Results show that the air-sea interaction, with a dominant role of LH flux, plays a major role in the development of nearly half (44%) of the MHWs in the Mediterranean Sea. This finding suggests*

*that oceanic processes play a key role during 56% of the onset cases. A further weakened role of heat flux is found during decline periods (being the major contributor in only 17% of declines), indicating that MHWs decay is also primarily driven by oceanic processes.*

[Line 186-188]. This sentence is a repetition of what already stated above. I suggest to rephrase or remove it.

Thank you for noting this. Our intention was to introduce to the reader the information merged in Fig. 3, briefly relating its content with the previous results.

We suggest rephrasing lines 187-190 as follows:

*Whereas during onset, MHWs are largely driven by heat flux exchange and most of them are accompanied by mixed layer shoaling, there are onset cases where MLDA are strongly positively correlated with SSTA, indicating surface warming evolves while the mixed layer deepens (Fig. 3a-top).*

The revised sentence provides additional information on the already discussed results for onset periods, reporting the existence of cases with positive heat-flux contribution and mixed layer deepening.

[Line 192]. "a significant MLDA-SSTA correlation is absent…". What do you mean with this sentence? To me is not clear.

Thank you for your comment. In Fig. 3a, the correlation coefficient (CC) values of the x axis range from -1 up to 1. The highest concentration of events in the bottom panel of Fig. 3a is observed in the right area of the heatmap, corresponding to large positive correlation (CC close to 1) between the MLD and SST time-series during declines (i.e., SST decrease and MLD decrease). However, looking at the entire bottom panel of Fig. 3a we see that all different cases exist, including cases with very small (non-significant) positive or negative CC values. We therefore report (Line 192) that apart from the most commonly observed finding for declines, there are several cases with no significant MLD-SST correlation.

[Line 248]. "…and oceanic processes". I would suggest to substitute oceanic processes with mixed layer heat budget analysis (or something equivalent).

Thank you for this suggestion. The part " and oceanic processes" has been entirely removed from the sentence.

**Comments on the example lines mentioned by the Reviewer in General Comments:**

**Lines 12-13:** Answer included in specific comment for these lines

**Lines 231-233 and Lines 265-268**

Our results show that the SST decrease during decline is primarily driven by air-sea heat flux in only 17% of the declines. On top of this, during most declines, a continuation of the mixed layer shoaling is found, suggesting that, in these cases, the SST decrease is not driven by mixing in the vertical. Based on the above, we assume that the surface cooling right after the peak intensity day

of most events is probably due to heat advection. In turn, the significant increase in MLD after the MHW end day found in most cases suggests that vertical mixing becomes important for the MHW decay after the end of the decline phase.

The contribution of oceanic advection mentioned in Lines 231-233 and Lines 265-268 is deduced from our analysis and is not investigated. For this reason, we briefly discuss this factor solely in terms of its potential role right after the peak-intensity day, as implied from our actually quantified results (i.e., the contribution of surface heat flux), and the enhanced/suppressed vertical mixing as indicated from MLD increase/decrease.

Our comment that this hypothesis applies especially for shorter decline periods (Line 268) is based on the higher SST-MLD correlation found for decline phases of shorter durations. This suggests that the progressively decreasing MLD during decline is more common during shorter declines. Considering also the weak contribution of air-sea heat flux during most declines (found to be weaker for shorter declines), this suggests a higher probability of advection being responsible for the observed surface cooling in these cases.

**Lines 227-229:**

The calculated percentage change in the MLD between the onset and the selected post-decline period (shown in histogram of Fig. 3c) suggests that a significant mixed layer deepening occurs after the MHW end day in most cases. We therefore believe that this finding supports the conclusion reported in the discussed lines "…*suggesting that vertical mixing eventually contributes to the heat dissipation.*" Let it be noted here that the examined post-decline period starts at the end day of the event, which means that SST will continue to decrease (by definition, as the end-day of the event is the last day when SST is above threshold).

**Changes in the manuscript:**

We have revised the manuscript, according to the suggestions and comments of the reviewers. All modifications made throughout the manuscript are visible in the version that includes tracked changes. The few re-wording suggestions (e.g., fluxes → flux) have been included throughout the text (and are visible in the manuscript that includes tracked changes) but are not individually listed here. Here we note the specific lines where content changes have been implemented. Please, note that references to lines are based on the revised document's line numbering.

**Abstract:**

Lines 12-14: These sentences have been rephrased to focus on results for air-sea heat flux, following the suggestion of Reviewer 2.

**Introduction:**

Line 45: Addition of reference (Reviewer 1).

Line 52: Non-peer reviewed reference has been removed (Reviewer 1).

**Data and methods:**

Line 58-9 and 63: Additional information on datasets has been included (Reviewer 1).

Line 69-70, 72: Additional information on the computation of climatology and thresholds has been included (Reviewer 2).

Line 88-92: A short paragraph has been added for our approach on applying a simplified heat budget equation (Reviewer 1).

Line 100-102: A sentence has been added clarifying that the contribution of heat flux is assessed in relation to a single residual term representing the cumulative effect of all oceanic factors influencing the SST tendency during a MHW phase (Reviewer 2).

Table 1: A different order is now used in the datasets presentation (Reviewer 1).

**Results:**

Line 125-127: This sentence has been revised for a more accurate description of Figs 1a,b (Reviewer 2)

Line 132-137: Trend values for the entire basin (with confidence intervals) have been added in this paragraph for frequency, mean intensity and duration (Reviewer 2)

Figure 1 has been replaced. The updated figure includes: different color palette, latitude (longitude) labels only for the left-hand (bottom) panels, change of deg → degC (Reviewer 1) and a switched criterion for statistical significance with black dots over the non-significant pixel values (Reviewer 2)

Figure 2 has been replaced. The updated figure includes: Change in colors of Fig1a and Fig.1b for onset and decline (Reviewer 1) and a typo correction in the y-label of Fig. 1c.

Line 176-179: Sentences have been rephrased to put more focus on the role of air-sea heat flux (Reviewer 2)

Line 197-199: Sentence has been rephrased to avoid repetitions (Reviewer 2)

Figure 4 has been replaced. The updated figure includes changes in colors for onset and decline (Reviewer 1)

**Conclusions:**

Line 258-259: Sentences have been merged (Reviewer 1)

Line 262, and 268-270: Changes have been introduced in this summary paragraph so that emphasis is put on the actually quantified results for the air-sea heat flux (Reviewer 2).

Line 286-291: Discussion on limitations of methods used for studying drivers (and relevant recommendations) has been enriched (Reviewer 1).

**References:**

Additions: Marullo et al. (2023), Paulson and Simpson (1977)

Removed: Darmaraki et al. (2023) (non-peer reviewed)

Correction: Year (2024) has been added in the references: McAdam et al. (2024) and Pirro et al. (2024). These papers were submitted in the context of the Ocean State Report #8.